# MedSimSearch: Sim2Real Agentic Learning for Medical Visual Reasoning

## Abstract

Developing autonomous agents for complex Medical Visual Reasoning is a critical goal, yet training them in real-world clinical settings is largely infeasible due to severe privacy, data, and safety constraints. While retrieval-augmented methods exist, they often depend on impractical multimodal indexing or fail to address the core challenge of learning interactive policies without real-world exposure. To bridge this gap, we introduce MedSimSearch, a novel framework based on Sim2Real Agentic Learning. The core innovation lies in leveraging a generative large multimodal model (LMM) to create a high-fidelity simulated retrieval environment. Within this safe, text-only simulation, our agent learns a robust search and reasoning policy, eliminating the need for multimodal data indexing while preserving patient privacy. To validate our approach, we evaluate the agent trained in simulation on realistic medical benchmarks using a curated private text corpus. Extensive experiments on VQAMed2019 and OmniMedVQA demonstrate that MedSimSearch significantly surpasses strong retrieval-augmented generation (RAG) baselines and shows enhanced robustness against hallucinations, paving a viable path for deploying trustworthy medical AI agents.[1]

## 1 Introduction

Developing AI agents for Medical Visual Reasoning represents a pivotal goal for supporting clinical diagnosis (Hu et al., 2024). However, transitioning these agents from laboratory settings to real-world clinical environments encounters a fundamental bottleneck: training them via Reinforcement Learning (RL) to autonomously interact with external knowledge systems under stringent medical constraints. The efficacy of RL hinges on extensive trial-and-error, yet the unique challenges of the medical domain render such agentic learning environments virtually inaccessible. High-quality multimodal data is sparse due to high costs and privacy regulations. Medical trajectory data is scarce, and the limited availability of expert annotators restricts supervisory signals. Additionally, the medical field lacks freely accessible retrieval tools, and direct RL training on real systems incurs prohibitive costs and privacy risks. This confluence of factors renders the development of an effective interactive agent a pressing unresolved challenge.

Existing approaches like Retrieval-Augmented Generation (RAG)(Lewis et al., 2020) fail to address the core difficulty of interaction-based policy learning. For instance, Search-R1(Jin et al., 2025b) is ill-suited for the multimodal demands of medical reasoning. Similarly, MMSearch-R1 (Jiang et al., 2024; Wu et al., 2025b), while learning a policy, relies on impractical image-text joint indexing. Moreover, traditional RAG's dependency on static knowledge bases exacerbates model hallucinations and operational barriers. These limitations underscore the urgent need for a privacy-compliant, efficient RL training mechanism tailored for medical agents.

Recent advancements in Large Language Models (LLMs)(OpenAI, 2024; Touvron et al., 2023) as executable "world models" offer a path forward. By simulating complex environments, LLMs enable the powerful Sim2Real paradigm: training agents in a controlled simulation without requiring real-world exposure(Zala et al., 2024; Corecco et al., 2024). In the context of search, this approach has been validated for data-scarce domains (Sun et al., 2025). In medical applications, this paradigm allows using LLMs to generate pseudo-documents that emulate expert knowledge, thereby facilitating RL training in a secure setting.

---

[1]We provide a statement on the use of LLMs in **Appendix A.1**

Motivated by these insights, we introduce MedSimSearch, a novel framework for Sim2Real Agentic Learning that employs a generative Large Multimodal Model (LMM) to simulate a retrieval environment. The framework's innovation lies in leveraging the LMM to produce high-quality pseudo-documents, eliminating reliance on multimodal indexing and external systems while ensuring privacy. Specifically, we implement a curriculum-based rollout simulation and optimize the agent's policy using Group Relative Policy Optimization (GRPO). To validate our approach, we curate a specialized medical text corpus. Extensive experiments on VQAMed2019 and OmniMedVQA demonstrate that MedSimSearch outperforms conventional RAG and strong baselines, achieving up to 73.26% accuracy on OmniMedVQA and advancing the practical deployment of autonomous medical reasoning systems.

In summary, our key contributions are:

- We propose MedSimSearch, the first framework to our knowledge that operationalizes the Sim2Real paradigm for agentic learning in the context of medical visual reasoning, addressing key challenges of privacy and data scarcity.

- We introduce a novel LMM-based simulation mechanism that generates high-quality pseudo-documents with a curriculum, creating a safe and effective environment for training a text-only medical search agent.

- Through extensive experiments, we demonstrate that our agent, trained entirely in simulation, significantly surpasses strong RAG and RL baselines on real-world benchmarks, validating the effectiveness of our Sim2Real approach.

## 2 RELATED WORK

### 2.1 MULTIMODAL LARGE LANGUAGE MODELS FOR MEDICAL VQA

In recent years, multimodal large language models (MLLMs) have significantly advanced the field of visual question answering (VQA) by integrating visual and textual features, leveraging the robust capabilities of LLMs such as GPT and LLaMA. Flamingo (Alayrac et al., 2022) incorporates cross-attention layers to fuse visual features, while BLIP2 (Li et al., 2023c) employs a novel Q-former to align pre-trained visual encoders with LLMs. InstructBLIP (Dai et al., 2023b) and LLaVA (Liu et al., 2023) further enhance performance through high-quality multimodal data and instruction fine-tuning. ALLVA (Chen et al., 2024) demonstrates that even smaller models (3B parameters) can achieve remarkable results with high-quality VQA data, underscoring the importance of data quality. In the medical domain, inspired by the success of medical LLMs such as ChatDoctor (Li et al., 2023d), MedicalGPT (Xu, 2023), HuatuoGPT (Zhang et al., 2023a; Chen et al., 2023), and Apollo (Wang et al., 2024b), researchers have developed models like Med-Flamingo (Moor et al., 2023), pre-trained on medical multimodal data, and LLaVA-Med (Li et al., 2023a), optimized using PubMed image-text pairs and synthetic VQA datasets. Zhang et al. (2023b) utilized PMC-OA (Lin et al., 2023) data to create the PMC-VQA dataset, enabling the development of the MedVInT model, while RadFM (Wu et al., 2025a) integrates 2D and 3D radiology images to build a radiology-focused multimodal model. However, Hu et al. (2024) note that current medical multimodal models still lag behind general medical models in specialized tasks, indicating the need for further technical advancements.

### 2.2 RETRIEVAL-AUGMENTED GENERATION FOR MEDICAL VQA

Retrieval-Augmented Generation (RAG) integrates external knowledge into LLMs to boost generation accuracy and mitigate hallucinations, as introduced by (Lewis et al., 2020). Initial RAG approaches relied on prompt-driven techniques, employing query formulation, decomposition, and iterative retrieval (Yu et al., 2023; Press et al., 2023; Shi et al., 2024; Jiang et al., 2023; Yoran et al., 2023), but these demanded sophisticated prompt design and robust model reasoning. To address these challenges, newer methods focus on supervised fine-tuning of smaller LLMs. For example, Self-RAG (Asai et al., 2023) leverages a self-reflective mechanism to iteratively improve outputs, while RetroLLM (Li et al., 2025) uses constrained decoding to extract precise evidence directly from corpora. Advanced test-time strategies, such as Monte Carlo Tree Search (MCTS), further enhance performance; RAG-star (Jiang et al., 2025) incorporates retrieved data into tree-based reasoning,

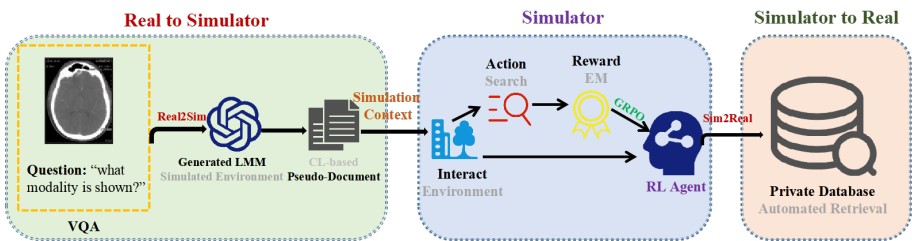

Figure 1: Overview of MedSimSearch.

and AirRAG (Feng et al., 2025) employs MCTS to unlock deeper reasoning and expand solution spaces. However, these methods often incur high computational costs, limiting practical deployment. In medical visual question answering (MedVQA), RAG significantly improves multimodal task performance by accessing public or proprietary databases. MedGraphRAG(Wu et al., 2025c) enhances LLMs in healthcare through hybrid semantic document splitting, entity extraction, and hierarchical knowledge graph construction. Nevertheless, traditional RAG is often inefficient, costly, and struggles to adapt to new diseases or real-time medical updates. While Search-R1 (Jin et al., 2025b;a) enables dynamic text retrieval, it is limited to single-modality data and untested in medical contexts. Concurrently, MMSearch-R1 (Jiang et al., 2024; Wu et al., 2025b) explores multimodal retrieval but depends on costly and privacy-sensitive image-text indexing. Most closely related to our approach is ZeroSearch (Sun et al., 2025), which pioneers the use of an LLM to simulate a search environment for training retrieval strategies. However, ZeroSearch is designed for general-domain web search and does not address the unique challenges of medical VQA, such as the need for precise, domain-specific knowledge retrieval and reasoning with clinical images.

Our work, MedSimSearch, distinctively addresses these gaps. Instead of improving RAG within the existing paradigm, we propose a shift to Sim2Real Agentic Learning. We introduce an RL framework that learns a robust search policy not by interacting with a real (and inaccessible) environment, but within a high-fidelity simulated environment generated by an LMM. This approach, while inspired by the simulation concept in works like ZeroSearch, is fundamentally different in its application to complex, multimodal medical reasoning and its focus on training an autonomous agent. By leveraging a curriculum-based simulation, MedSimSearch learns an effective, text-only retrieval policy, eliminating the need for multimodal indexing and ensuring privacy compliance, thus making it both practical for clinical settings and effective for complex medical reasoning.

## 3 MEDSIMSEARCH

In this section, we present the detailed design of MedSimSearch, a novel framework for Sim2Real agentic learning tailored for medical visual reasoning. Figure 1 presents the Overview of our proposed MedSimSearch. Our approach circumvents the challenges of real-world interaction by leveraging a generative LMM to create a high-fidelity simulated environment. Within this simulation, an RL agent learns a robust, text-only search policy without requiring impractical multimodal indexing, thus ensuring privacy compliance and scalability. The input consists of a medical reasoning task, defined by a question $x$ and an associated image $i$. The agent processes this input to generate text queries and interact with the simulated environment. We describe: (1) the Sim2Real RL framework, including adaptations of GRPO; (2) the simulated environment design via pseudo-document generation; (3) the agent's training template; and (4) the reward model design.

**GRPO with Pseudo-Document Generation.** GRPO Shao et al. (2024) improves policy optimization stability:

$$
\begin{aligned}
J_{\text{GRPO}}(\theta) =& \mathbb{E}_{(x,i)\sim\mathcal{D},\{y^j\}_{j=1}^G\sim\pi_{\text{old}}(\cdot|x,i;\mathcal{P})} \Bigg[ \frac{1}{G}\sum_{j=1}^G \frac{1}{|y^j|}\sum_{t=1}^{|y^j|} I(y_t^j)\min\Bigg( \frac{\pi_\theta(y_t^j|x,i,y_{<t}^j;\mathcal{P})}{\pi_{\text{old}}(y_t^j|x,i,y_{<t}^j;\mathcal{P})}\hat{A}_t^j, \\
& \text{clip}\left(\frac{\pi_\theta(y_t^j|x,i,y_{<t}^j;\mathcal{P})}{\pi_{\text{old}}(y_t^j|x,i,y_{<t}^j;\mathcal{P})},1-\epsilon,1+\epsilon\right)\hat{A}_t^j \Bigg) - \beta D_{\text{KL}}[\pi_\theta\|\pi_{\text{ref}}] \Bigg],
\end{aligned}
$$

(1)

---

**Algorithm 1** MedSimSearch Response Rollout with Pseudo-Document Generation

---

**Require:** Input question $x$, image $i$, policy model $\pi_\theta$, medical text corpus $\mathcal{C}$, max action budget $B$
**Ensure:** Final response $y$
1: Initialize rollout sequence $y \leftarrow \emptyset$
2: Initialize action count $b \leftarrow 0$
3: **while** $b < B$ **do**
4:     Initialize current action rollout $y_b \leftarrow \emptyset$
5:     **while** True **do**
6:         Generate token $y_t \sim \pi_\theta(\cdot|x, i, y + y_b)$
7:         Append $y_t$ to $y_b \leftarrow y_b + y_t$
8:         **if** $y_t \in \{\langle/\text{info}\rangle, \langle/\text{answer}\rangle, \langle\text{eos}\rangle\}$ **then**
9:             **break**
10:         **end if**
11:     **end while**
12:     Update $y \leftarrow y + y_b$
13:     **if** $\langle\text{info}\rangle\langle/\text{info}\rangle$ detected in $y_b$ **then**
14:         Extract query context $q \leftarrow \text{Parse}(y_b, \langle\text{think}\rangle, \langle/\text{think}\rangle)$
15:         Generate pseudo-document $\mathcal{P} \leftarrow \pi_\theta(\cdot|q, i)$ or retrieve from $\mathcal{C}(q)$
16:         Append $\mathcal{P}$ to rollout $y \leftarrow y + \langle\text{info}\rangle\mathcal{P}\langle/\text{info}\rangle$
17:     **else if** $\langle\text{answer}\rangle\langle/\text{answer}\rangle$ detected in $y_b$ **then**
18:         **return** final response $y$
19:     **else**
20:         Append rethink prompt $y \leftarrow y +$ "Rethink required."
21:     **end if**
22:     Increment action count $b \leftarrow b + 1$
23: **end while**
24: **return** final response $y$

---

Table 1: Template for Curriculum Pseudo-Document Simulation in MedSimSearch. The `[useful/ noisy]` keyword controls document quality through zero-shot prompting.

---

**Prompt Template**

You are a medical knowledge simulator. Given a query, you need to generate five `[useful/ noisy]` pseudo-documents for the medical VQA question: `[question]` with associated image: `[image description]` whose ground truth answer is `[ground truth]`. Each pseudo-document should contain about 30 words, and these documents should contain `[useful/ noisy]` medical information relevant to the question and image. Query: `[query]` Output:

---

where $\epsilon$ and $\beta$ are hyperparameters, and $\hat{A}_t^j$ are advantages computed from relative rewards within the group.

### 3.1 PSEUDO-DOCUMENT GENERATION WITH CURRICULUM LEARNING

MedSimSearch employs a generative LMM to produce pseudo-documents through a curriculum-based rollout simulation. The key innovation is a zero-shot prompting strategy that dynamically controls the difficulty of the simulated retrieval environment without fine-tuning the LMM. This is achieved by instructing the LMM to generate a mixture of "useful" (accurate and relevant) and "noisy" (misleading or irrelevant) pseudo-documents during its interaction with the RL agent. Our approach presents a distinct path from ZeroSearch (Sun et al., 2025), which implements the curriculum learning paradigm through model training. In contrast, our method pursues a more straightforward and efficient alternative, achieving significant gains in simplicity and operational speed.

The mixing ratio follows a predefined curriculum schedule: initially, the simulation provides mostly useful documents (e.g., 80% useful, 20% noisy), easing the agent into learning the retrieval policy. As training progresses, the proportion of noisy documents is gradually increased (e.g., to 20% useful, 80% noisy), challenging the agent to develop robust discrimination capabilities. The `[useful/`

Table 2: MedSimSearch Training Template Structure

| Component | Description |
|---|---|
| Instruction | Answer the given medical VQA question using the provided image while maintaining data privacy constraints. Generate search queries if needed. |
| Reasoning Phase | Conduct analytical reasoning on the image and question within `<think>` and `</think>` tags. |
| Knowledge Generation | If additional medical knowledge is required, generate a text query within `<search>` and `</search>` tags to produce pseudo-documents, enclosed in `<info>` and `</info>` tags. |
| Final Answer | Provide the conclusive answer within `<answer>` and `</answer>` tags. |
| Input Format | Question: `{specific medical question}` Image: `{image description or embedding}` |

`noisy]` keyword in the prompt template (Table 1) is programmatically controlled to implement this schedule. The entire process is zero-shot; the LMM itself is not trained but is used as a controllable simulator to generate documents of varying quality on the fly.

Given a medical VQA input $(x, i)$, the LMM generates a set of pseudo-documents $\mathcal{P}$ conditioned on the current curriculum step:

$$\mathcal{P} \sim \pi_{\text{LMM}}(\cdot|x, i; \text{prompt}_{\text{curriculum}}), \qquad (2)$$

where the prompt instructs the LMM to synthesize a curriculum-adjusted mixture of evidence. (See Appendix A.3 for detailed prompt designs and scheduling algorithm). These pseudo-documents are enclosed within special tokens ⟨info⟩ and ⟨/info⟩, serving as context for reasoning while maintaining text-only retrieval to avoid multimodal indexing.

For validation in private-data scenarios, we curate a specialized medical text corpus $\mathcal{C}$, comprising anonymized clinical texts and LMM-synthesized records to simulate a private knowledge base. (See Appendix A.4 for construction details). The overall pipeline is illustrated in Algorithm 1.

### 3.2 TRAINING TEMPLATE

The training template structures the LMM's output into iterative phases: reasoning based on the image and question, pseudo-document generation via text query, and final answer. This template ensures flexibility while maintaining structural consistency across medical domains, as detailed in Table 2.

### 3.3 REWARD MODEL DESIGN

The reward function based on a rule-based reward system that consists solely of final outcome rewards, which assess the correctness of the model's response.

$$r_\phi((x, i), y) = \alpha \cdot \text{EM}(a_{\text{pred}}, a_{\text{gold}}) \qquad (3)$$

where $\text{EM}(a_{\text{pred}}, a_{\text{gold}})$ refers to Exact match score between predicted and ground truth answers which followed by Jin et al. (2025b).

## 4 EXPERIMENTS

### 4.1 DATA AND EVALUATION

We evaluate MedSimSearch on two comprehensive medical visual question answering benchmarks: (1) VQAMed2019 (Ben Abacha et al., 2019), which includes 3,200 medical images with 12,792 QA pairs in the training set, 500 medical images with 2,000 QA pairs in the validation set, and 500

medical images with 500 QA pairs in the test set, covering a range of medical imaging tasks; and (2) OmniMedVQA (Hu et al., 2024), a large-scale, multi-specialty medical VQA benchmark containing over 300,000 triplets across radiology, dermatology, pathology, and ophthalmology, with diverse reasoning types (e.g., diagnosis, procedural, descriptive). We use the official train/val/test splits for all experiments. In line with the methodology proposed by Hu et al. (2024), we employ Exact Match (EM) as the evaluation metric for the OmniMedVQA dataset. A prediction is considered correct if its normalized representation precisely corresponds to any of the normalized ground-truth answers. For the VQAMed2019 dataset, we adhere to the evaluation framework outlined by Ben Abacha et al. (2019), utilizing two primary metrics: Accuracy and BLEU.

Following Hu et al. (2024), we adopt Exact Match (EM) as our evaluation metric for OmniMedVQA dataset. A prediction is deemed correct if its normalized form exactly matches any of the normalized ground-truth answers. To evaluate VQAMed2019 dataset, we follow Ben Abacha et al. (2019) based on two primary metrics: Accuracy and BLEU.

## 4.2 EXPERIMENTAL SETUP

We conduct experiments using the Qwen2.5-VL-3B model (Qwen et al., 2025; Bai et al., 2025), a multimodal large language model (MLMM). To simulate real-world medical text retrieval scenarios, we employ GPT-4o (OpenAI, 2024) as the external LMM to generate pseudo-documents and simulate a retrieval-augmented large model. During evaluation, all methods utilize GPT-4o for pseudo-document generation to ensure a fair comparison. The number of generated pseudo-documents is fixed at five across all methods for consistency. To validate our approach in realistic private-data scenarios, we curate a specialized medical text corpus $\mathcal{C}$, comprising anonymized clinical texts and LMM-synthesized medical records, which serves as a private database for text retrieval. We evaluate MedSimSearch on two medical VQA datasets: VQAMed2019 and OmniMedVQA, using their official train/val/test splits to assess performance. We use the training sets of VQAMed2019 and OmniMedVQA to create dataset for all RL-based approaches, respectively. For prompt-based baselines, we use the Instruct variant of Qwen2.5-VL-3B, as Base models often struggle with task-specific instructions. For RL-based methods, we evaluate both Base and Instruct variants to assess generality across model types. We experiment with two RL algorithms to train MedSimSearch: Proximal Policy Optimization (PPO), and GRPO. Unless otherwise specified, PPO is used as the default training algorithm. During experimentation, we deploy the simulation server on 4 NVIDIA A100 GPUs and conduct RL training on another 4 NVIDIA A100 GPUs. During inference, all models interact with the curated medical text corpus $\mathcal{C}$ via GPT-4o-generated queries to ensure a fair and privacy-compliant comparison. For further implementation details, including complete hyperparameter configurations, please refer to Appendix A.2.

## 4.3 BASELINES

In evaluating the performance of our proposed MedSimSearch framework, we conduct a thorough comparison against a diverse array of baseline models for the OmniMedVQA dataset, encompassing both zero-shot and fine-tuned paradigms to provide a comprehensive assessment across different capabilities and domains. For zero-shot general-purpose vision-language models (VLMs), we assessed the performance ceiling of large-scale models without domain-specific tuning, including BLIP-2 (Li et al., 2023b), InstructBLIP (Dai et al., 2023a), LLaVA (Liu et al., 2023), and MiniGPT-4 Zhu et al. (2023), alongside more recent advancements such as Qwen2-VL-2B/7B (Wang et al., 2024a), and their enhanced iterations, Qwen2.5-VL-3B/7B (Bai et al., 2023). Additionally, we incorporated zero-shot medical VLMs trained on biomedical data, such as LLaVA-Med (Li et al., 2023a), RadFM (Wu et al., 2025a), Med-Flamingo (Moor et al., 2023), and MedVInT (Zhang et al., 2024), to establish benchmarks for domain-specific performance without further adaptation. For fine-tuned VLMs, we utilized models trained on the OmniMedVQA training split, including supervised fine-tuning (SFT) baselines where robust general-purpose models like Qwen2-VL-2B and Qwen2.5-VL-3B were optimized using standard instruction tuning, as well as results from RAG-enhanced versions of Qwen2.5-VL-3B and Qwen2.5-VL-7B. Furthermore, we included RL-based baselines, notably the state-of-the-art reward-learning VLMs Med-R1 (Lai et al., 2025) in variants of 2B and 3B, with and without reasoning modules, to directly compare against MedSimSearch and evaluate its advancements in reinforcement learning strategies tailored for medical VQA tasks.

Table 3: Comprehensive evaluation on OmniMedVQA benchmark. Results are reported as exact match accuracy (%) across eight medical modalities. Our MedSimSearch framework achieves state-of-the-art performance (overall), outperforming all zero-shot, fine-tuned, RAG, and RL-based baselines. † denotes RL-based baselines. Best and second-best results are in **Bold** and underline, respectively.

| Type | Method | Modality | | | | | | | | overall |
| | | CT | MRI | X-Ray | US | Der | FP | OCT | Micro | |
|---|---|---|---|---|---|---|---|---|---|---|
| Zero-shot & SFT VLMs | BLIP-2 | 56.74 | 41.32 | 67.58 | 37.27 | 40.65 | 46.24 | 68.08 | 50.40 | 51.04 |
| | InstructBLIP | 28.72 | 33.15 | 61.04 | 41.25 | 62.22 | 50.31 | 42.59 | 46.29 | 45.70 |
| | LLaVA | 17.73 | 26.72 | 30.70 | 18.66 | 49.74 | 47.11 | 33.73 | 28.87 | 31.66 |
| | MiniGPT-4 | 22.81 | 27.48 | 38.30 | 25.50 | 40.25 | 38.33 | 31.40 | 36.23 | 32.54 |
| | LLaVA-Med | 18.69 | 27.47 | 30.68 | 29.88 | 44.95 | 39.03 | 34.61 | 33.29 | 32.33 |
| | RadFM | 27.56 | 24.06 | 30.95 | 16.57 | 39.21 | 36.86 | 32.80 | 27.97 | 29.50 |
| | Med-Flamingo | 31.28 | 26.34 | 44.01 | 31.69 | 48.56 | 41.26 | 25.16 | 30.03 | 34.29 |
| | MedVInT | 40.74 | 43.10 | 55.10 | 41.26 | 29.11 | 31.84 | 23.26 | 32.02 | 37.05 |
| | Qwen2-VL-2B | 45.10 | 38.57 | 39.32 | 30.86 | 35.83 | 43.17 | 35.14 | 36.85 | 38.11 |
| | Qwen2-VL-2B (SFT) | 51.74 | 52.83 | 65.57 | 47.65 | 51.91 | 52.26 | 53.99 | 56.58 | 54.07 |
| | Qwen2-VL-7B | 61.46 | 40.34 | 64.27 | 51.11 | 60.02 | 63.70 | 53.34 | 53.64 | 56.40 |
| | Qwen2.5-VL-3B | 53.87 | 54.23 | 61.84 | 32.69 | 52.94 | 62.47 | 56.23 | 59.64 | 54.24 |
| | Qwen2.5-VL-3B (SFT) | 56.06 | 60.81 | 69.23 | 41.77 | 60.11 | 69.19 | 63.95 | 65.66 | 60.85 |
| | Qwen2.5-VL-7B | 60.44 | 58.44 | 73.99 | 50.46 | 62.48 | 67.66 | 67.40 | 61.87 | 60.29 |
| Search & RL | Qwen2.5-VL-3B (RAG) | 54.91 | 57.86 | 65.53 | 37.42 | 56.45 | 67.61 | 59.48 | 62.25 | 57.69 |
| | Qwen2.5-VL-7B (RAG) | 62.44 | 64.52 | 69.98 | 49.24 | 65.83 | 70.12 | 71.24 | 68.03 | 65.18 |
| | Med-R1-2B (Think)† | 66.30 | 71.61 | 84.52 | 57.31 | 72.33 | 71.33 | 71.96 | 70.80 | 70.77 |
| | Med-R1-3B (Nothink)† | 69.89 | 72.91 | 84.52 | 43.91 | 73.62 | 80.10 | 84.18 | 71.44 | 72.57 |
| Ours | **MedSimSearch (GPT-4o)** | 68.86 | 71.56 | 85.83 | 48.67 | 74.35 | 81.23 | 83.55 | 72.01 | **73.26** |
| | **MedSimSearch (Database)** | 67.96 | 72.32 | 83.88 | 49.54 | 75.89 | 78.23 | 84.85 | 71.39 | 73.01 |

In contrast, for the VQAMed2019 dataset, we adopted a more focused comparison, aligning with the ImageCLEF 2019 (Ben Abacha et al., 2019) evaluation framework, to contrast MedSimSearch against a select group of models. This included zero-shot variants of Qwen2.5-VL-3B/7B, which provide a baseline for out-of-the-box performance on medical imaging tasks, as well as their RAG-enhanced counterparts (Qwen2.5-VL-3B/7B), which leverage external knowledge retrieval to boost accuracy. Additionally, we evaluated the SFT version of Qwen2.5-VL-3B, fine-tuned on the VQA-Med 2019 training data, to assess the impact of domain-specific adaptation. This streamlined set of baselines allows us to isolate the contributions of zero-shot generalization, retrieval augmentation, and supervised fine-tuning in the context of VQAMed2019's radiology-focused challenges, providing a direct comparison to MedSimSearch's innovative approach of simulated retrieval environments.

## 4.4 MAIN RESULTS

**Results on the OmniMedVQA Benchmark** The comprehensive evaluation on the OmniMed-VQA benchmark, detailed in Table 3, demonstrates that our proposed MedSimSearch framework achieves state-of-the-art performance. The primary variant, MedSimSearch (GPT-4o), attains a top overall accuracy of 73.26%. This result significantly outperforms strong retrieval-augmented generation (RAG) baselines, such as Qwen2.5-VL-7B (RAG) at 65.18%, by a margin of 8.08 percentage points. Furthermore, our method establishes a new benchmark by surpassing the previous state-of-the-art RL-based method, Med-R1-3B (72.57%). The robustness of our approach is underscored by

Table 4: Performance metrics (%) on the VQAMed2019 dataset, evaluated across overall accuracy, BLEU score, and category-specific results (Modality, Plane, Organ, Abnormality). **Bold** values indicate the best performance for each metric.

| Method | Category | | | | Overall | BLEU |
|---|---|---|---|---|---|---|
| | **Modality** | **Plane** | **Organ** | **Abnormality** | | |
| ImageCLEF2019 | 20.2 | 19.2 | 18.4 | 46.0 | 62.4 | 64.4 |
| Qwen2.5-VL-3B (zero-shot) | 15.3 | 14.8 | 10.5 | 35.4 | 41.9 | 46.4 |
| Qwen2.5-VL-7B (zero-shot) | 21.9 | 20.6 | 21.3 | 48.5 | 63.2 | 65.6 |
| Qwen2.5-VL-3B (RAG) | 19.9 | 20.2 | 16.5 | 46.4 | 61.8 | 63.9 |
| Qwen2.5-VL-7B (RAG) | 25.4 | 24.7 | 26.8 | 51.2 | 64.6 | 67.2 |
| Qwen2.5-VL-3B (SFT) | 26.1 | 25.3 | 24.23 | 51.7 | 64.9 | 68.4 |
| **MedSimSearch (GPT-4o)** | 28.6 | 27.6 | 26.8 | 54.6 | **66.8** | **74.3** |
| **MedSimSearch (Database)** | 28.1 | 26.9 | 26.2 | 55.3 | 66.4 | 73.7 |

its leading performance across key medical modalities, including X-Ray (85.83%), Fundus Photography (FP) (81.23%), and OCT (83.55%). An alternative configuration, MedSimSearch (Database), which utilizes a curated text corpus, also delivers exceptional performance at 73.01%. This confirms that the superiority stems from the learned retrieval policy itself, which is effective across different knowledge sources.In summary, the results conclusively show that MedSimSearch sets a new state-of-the-art for medical VQA by leveraging a dynamically learned retrieval strategy within a simulated environment.

**Results on the VQA-Med 2019 Dataset** As shown in Table 4, the proposed MedSimSearch framework achieves state-of-the-art performance on the VQA-Med 2019 benchmark. The primary variant, MedSimSearch (GPT-4o), attains the highest overall accuracy of 28.6%, surpassing the strongest baseline, Qwen2.5-VL-3B (SFT), by 2.5 percentage points. This superiority is consistent across all reported metrics. Our method also achieves the top BLEU score (27.6) and demonstrates robust performance across medical categories, most notably achieving a 74.3% accuracy in the diagnostically critical Abnormality category. The competitive results of the MedSimSearch (Database) variant (28.1% overall accuracy) further confirm the generalizability of the learned retrieval policy. These results conclusively demonstrate that MedSimSearch effectively generalizes to established medical VQA tasks, outperforming both standard fine-tuning and RAG-based approaches.

## 5 ANALYSIS

### 5.1 GENERALIZATION ACROSS DIFFERENT SIMULATOR LMMS

A critical question for our method is whether the learned retrieval policy overfits to the specific LMM used for simulation during training. To test this, we evaluated the fixed MedSimSearch agent on the OmniMedVQA test set, using a variety of unseen, off-the-shelf multimodal and text-only LMMs as the simulation environment. These models were accessed via their public APIs and were presented with the same input format (image+question for multimodal models; question-only for text-only models) to generate pseudo-documents. As shown in Figure 2, MedSimSearch's performance declines moderately as the simulation LMM's capability decreases, with accuracy dropping from 73.26% (GPT-4o) to 71.20% (LLaMA-3-70B) and 69.85% (Qwen-7B). The smooth and minor decline, retaining 97.2% performance even with a text-only LLaMA-3-70B, indicates the learned retrieval policy's robustness and generalizability, not being overly dependent on GPT-4o's specific knowledge. This suggests the policy effectively utilizes the quality of external knowledge and decouples retrieval strategy learning from specific LMM choices, offering flexibility for deployment under real-world constraints like cost and privacy.

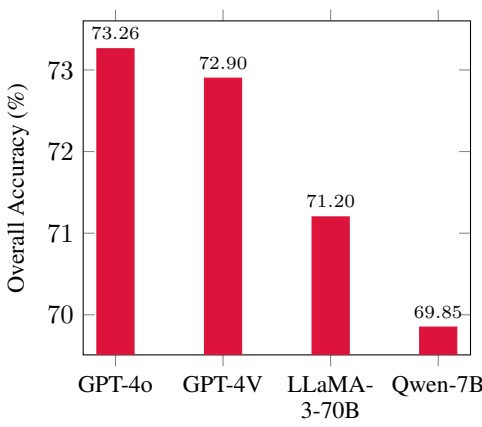 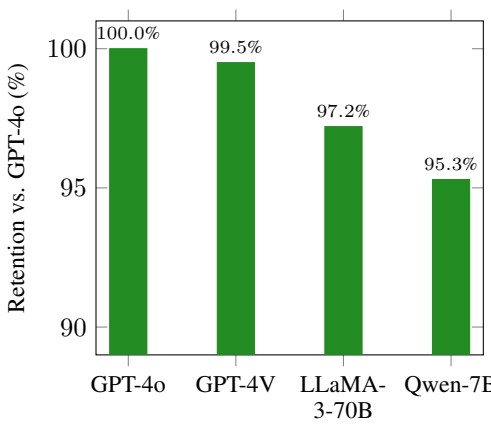

(a) Performance across different LLM simulators      (b) Performance retention relative to GPT-4o

Figure 2: Performance of MedSimSearch with different LLM simulators. The left plot shows absolute accuracy, while the right plot shows performance retention relative to GPT-4o. Results demonstrate that even when using a pure text-only model (LLaMA-3-70B), the performance retention remains high at 97.2%, validating the robustness of our approach.

## 6  CONCLUSION

In this work, we identify and address a critical bottleneck in deploying RL for Medical Visual Question Answering (MedVQA): the impracticality of training RL agents to interact with knowledge retrieval systems in real-world clinical settings due to data sparsity, privacy constraints, and the lack of accessible medical search tools. To overcome this, we introduced MedSimSearch, a novel framework that leverages a generative LMM as a simulated retrieval environment. By generating high-quality pseudo-documents, our method eliminates the need for multimodal indexing and interaction with real-world retrieval systems during training. Extensive experiments on the MedVQA2019 and OmniMedVQA benchmarks demonstrate that MedSimSearch not only achieves state-of-the-art performance but also significantly outperforms conventional RAG baselines. Our work provides a effective, privacy-compliant, and scalable pathway for training interactive AI systems in data-sensitive domains, marking a meaningful step towards the practical clinical deployment of advanced MedVQA models.

### LIMITATIONS AND FUTURE WORK

Despite its promising results, MedSimSearch has certain limitations that open avenues for future research. First, the quality of the generated pseudo-documents is inherently dependent on the knowledge embedded within the external LMM (e.g., GPT-4o). This may lead to potential inaccuracies or hallucinations in the simulated content, which could mislead the RL agent during training. Future work could explore integrating verifiable medical knowledge graphs or implementing a fact-checking module to enhance the reliability of the generated context. Furthermore, exploring methods to quantify the "reality gap" between the simulation and real-world scenarios would be crucial for understanding the transferability of learned policies. Second, our current framework focuses on simulating a text-only retrieval environment. While this is a strategic decision to address privacy and complexity, it does not fully capture the challenge of retrieving and reasoning with raw medical images. A promising direction is to extend the simulation to incorporate synthetic or perturbed medical images, creating a more comprehensive multimodal training environment. This would involve tackling the significant challenge of generating high-fidelity, diagnostically meaningful medical images. Finally, the rule-based reward function, while effective, is relatively simplistic. Designing a more nuanced reward model that incorporates clinical consensus or fine-grained reasoning verification could further refine the agent's decision-making process. We believe addressing these limitations will further bridge the gap between AI research and real-world clinical utility.

ETHICS STATEMENT

Our research on MedSimSearch prioritizes ethical considerations to address the sensitive nature of medical data and AI deployment in healthcare. We adhere to strict data privacy regulations, including anonymizing all clinical texts and synthesizing medical records using large multimodal model (LMMs) to avoid using real patient data, ensuring compliance with standards such as HIPAA and GDPR. The curated medical text corpus used for training and evaluation contains no personally identifiable information, and all experiments are conducted in a controlled, privacy-compliant environment on secure computational resources. We acknowledge the potential risks of model hallucinations and biases in pseudo-document generation, mitigating these through rigorous validation against established benchmarks (e.g., VQAMed2019, OmniMedVQA) and transparent reporting of performance metrics. Our work aims to enhance clinical decision-making without replacing human judgment, and we commit to ongoing evaluation to prevent unintended societal or clinical harms. We have not conducted human subject research, and no external human feedback was solicited beyond the anonymized dataset creation process, which was overseen by institutional guidelines.

REPRODICIBILITY STATEMENT

To facilitate reproducibility of our work, we provide detailed descriptions of the model architecture, training methodology, and experimental setup throughout the paper:

- **Model Description:** The MedSimSearch framework is introduced in **Section 3**, including the reinforcement learning pipeline, pseudo-document generation mechanism, and curriculum-based training strategy.
- **Data Description:** The construction and composition of the medical text corpus used for training and evaluation are detailed in **Appendix A.4**. Additionally, the datasets used for evaluation (VQAMed2019 and OmniMedVQA) are described in **Section 4.1**.
- **Parameter Settings:** Key hyperparameters and training configurations, such as learning rates, RL algorithm settings, and prompt templates, are provided in **Appendix A.2** and **Appendix A.3**.

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

# A APPENDIX

## A.1 USE OF LLMs

**Use of LLMs in Paper Writing Process**  In the preparation of this manuscript, we utilize large language models (LLMs) to assist with language refinement and translation to enhance clarity and readability for an international audience. Specifically, LLMs are employed for tasks such as grammar correction, sentence rephrasing, and translating technical content from Chinese to English where applicable. These tools, including models like Gpt-4o and deepseek, are used to polish sections of the text, such as the abstract, introduction, and experimental descriptions, ensuring adherence to academic writing standards. However, the core scientific content, including research design, experimental results, and analysis, was independently developed by the authors. All LLM-assisted edits are reviewed and approved by the research team to maintain the integrity and originality of the work. No LLMs were used to generate novel ideas, hypotheses, or substantive intellectual contributions.

## A.2 EXPERIMENTAL CONFIGURATION AND DETAILS

For the experiments conducted on Qwen2.5-VL-3B, we employ a lightweight supervised fine-tuning (SFT) to train the simulation LMM, using Qwen2.5-3B-Instruct as the backbone. The learning rate for this SFT is set to $1 \times 10^{-6}$. For the MedSimSearch framework, we adopt the Group Relative Policy Optimization (GRPO) reinforcement learning algorithm for training. In the GRPO setting, the policy LMM is trained with a learning rate of $1 \times 10^{-6}$, and five responses are sampled per prompt. The initial noise probability $p_s$ is set to 0, and the final noise probability $p_e$ is set to 0.25 for Qwen2.5-VL-3B. Additional experimental details are referenced from Sun et al. (2025).

### A.3 Zero-Shot Curriculum Implementation Details

This section provides the technical details of the zero-shot curriculum learning strategy outlined in Section 3.1.

#### A.3.1 Dynamic Prompting for Difficulty Control

The prompts in Table 5 are used zero-shot by the LMM. The `[quality]` placeholder is dynamically replaced with `useful` or `noisy` based on the current curriculum step to generate the corresponding document type without any model fine-tuning.

Table 5: Zero-Shot Prompt Templates for Dynamic Difficulty Generation

| Document Type | Prompt Template |
|---|---|
| **Useful** | You are a medical knowledge simulator. Given a query, you need to generate five **useful** pseudo-documents for the medical VQA question: [question] with associated image: [image description]. Each document should contain about 30 words, and provide **accurate and directly relevant** medical information that helps answer the question. |
| **Noisy** | You are a medical knowledge simulator. Given a query, you need to generate five **noisy** pseudo-documents for the medical VQA question: [question] with associated image: [image description]. Include **misleading or partially incorrect** information such as: incorrect anatomical locations, plausible but unrelated symptoms, or reversed causal relationships. |

#### A.3.2 Curriculum Scheduling Algorithm

The probability $p_{\text{noise}}$ of generating a batch of *noisy* documents (as opposed to *useful* ones) follows a linear schedule over training iterations $t$:

$$p_{\text{noise}}(t) = p_{\text{start}} + (p_{\text{end}} - p_{\text{start}}) \times \frac{t}{T_{\text{total}}} \tag{4}$$

where $p_{\text{start}} = 0.2$, $p_{\text{end}} = 0.8$, and $T_{\text{total}}$ is the total number of training iterations. This provides a smooth increase in difficulty for the RL agent.

### A.4 Construction of the Medical Text Corpus $\mathcal{C}$

This section details the construction process of the specialized medical text corpus $\mathcal{C}$ used for validation in private-data scenarios, as referenced in Section 3.1.

#### A.4.1 Corpus Composition and Sourcing

The corpus $\mathcal{C}$ is a pure text-based database designed to emulate a privacy-compliant clinical knowledge base. It consists of two components to ensure authenticity and coverage while adhering to privacy constraints:

- **Anonymized Clinical Texts from MIMIC-CXR:** Sourced exclusively from the MIMIC-CXR dataset Johnson et al. (2019), these include de-identified radiology reports and clinical notes. Anonymization is performed using a BERT-based NER model, followed by manual verification to remove all protected health information (PHI).

- **LMM-Synthesized Medical Records:** A smaller proportion of synthetic records is generated using GPT-4o in zero-shot mode, guided by structured prompts based on anonymized MIMIC-CXR data, to enhance diversity. These constitute approximately 10% of the corpus.

### A.4.2 CONSTRUCTION PIPELINE AND STATISTICS

The construction pipeline includes data cleaning, deduplication, and vector indexing for efficient retrieval. Table 6 summarizes the composition and scale of $\mathcal{C}$.

Table 6: Composition and Statistics of the Medical Text Corpus $\mathcal{C}$

| Component | Source | Number of Documents |
|---|---|---|
| Anonymized Radiology Reports | MIMIC-CXR | 50,000 |
| Synthetic Medical Records | GPT-4o | 5,000 |
| **Total** | | **55,000** |

The corpus contains approximately 55,000 document chunks, processed into self-contained semantic units suitable for retrieval.

### A.4.3 EXAMPLES OF CORPUS ENTRIES

The following examples illustrate the composition of $\mathcal{C}$. Example 7 shows a typical anonymized entry from MIMIC-CXR, while Example 8 demonstrates a synthesized entry used to augment the corpus.

Table 7: Example of Anonymized Report from MIMIC-CXR

**Source:** MIMIC-CXR
**Content:**
The lungs are clear. No focal consolidation, pneumothorax, or pleural effusion. The cardiomediastinal silhouette is normal in size and configuration. No acute bony abnormalities.

Table 8: Example of LMM-Synthesized Medical Record

**Source:** GPT-4o Synthesis
**Content:**
Chest radiograph demonstrates extensive bilateral interstitial opacities with a basilar predominance, suggestive of fibrotic lung disease. There is associated traction bronchiectasis. The cardiac silhouette is enlarged. Findings are most compatible with a diagnosis of idiopathic pulmonary fibrosis.

### A.4.4 INDEXING AND RETRIEVAL IMPLEMENTATION

The corpus is indexed using Facebook AI Similarity Search (FAISS) with a pre-trained biomedical sentence transformer model. This enables efficient similarity search during the retrieval simulation phase of MedSimSearch. The composition of $\mathcal{C}$—primarily grounded in real clinical data with minimal synthesis—ensures that our validation robustly tests the MedSimSearch framework under realistic and privacy-conscious conditions.

