# OpenReview forum: "MedSimSearch: Sim2Real Agentic Learning for Medical Visual Reasoning"
_ICLR.cc/2026/Conference — ICLR 2026 Conference Withdrawn Submission_

### Official Review · Reviewer_TA4o · 2025-10-29

**Soundness:** 3
**Presentation:** 3
**Contribution:** 2
**Rating:** 4
**Confidence:** 2

**Summary:**

The work introduces MedSimSearch, a framework that operationalizes the Sim2Real paradigm for agentic learning in the context of medical visual reasoning. The work is motivated by the real-world challenges of training autonomous agents in clinical settings, which are often constrained by privacy, data scarcity, and safety concerns. The core proposal is to leverage a generative Large Multimodal Model to create a text-only simulated retrieval environment. Within this safe simulation, an RL agent learns a robust, text-only search policy, eliminating the need for impractical multimodal data indexing. The agent's policy is optimized using GRPO through a curriculum-based rollout that progressively introduces noisy/negative pseudo-documents to improve robustness. The agent, trained entirely in this simulated environment, is then validated on real-world medical benchmarks using a private text corpus.

**Strengths:**

- The work is very well-motivated. It identifies a critical and practical bottleneck in deploying interactive medical AI: the inability to train agents on real systems due to privacy, access, and data scarcity. The proposed Sim2Real approach, using a text-only simulator, is a pragmatic and well-justified workaround for this significant hurdle.
- The work integrates several advanced methods to address this problem. The use of an LMM-as-simulator, combined with a curriculum learning strategy (dynamically mixing "useful" vs. "noisy" pseudo-documents), and optimization via GRPO, represents a novel and technically sound pipeline for this domain.
- The experiments are thorough and well-designed. The authors evaluate MedSimSearch on two benchmarks and compare against a comprehensive set of baselines, including zero-shot, SFT, and RAG models. The validation on a specialized, curated medical text corpus to test the Sim2Real transfer is a strong component of the evaluation. The results clearly demonstrate gains over existing SOTA methods.

**Weaknesses:**

- A major concern is the high computational cost of the proposed method. The paper states that training requires 4 NVIDIA A100 GPUs for the simulation server and another 4 A100 GPUs for the RL training. This 8-GPU setup makes the results difficult to reproduce for many research labs. Furthermore, the paper does not discuss the deployment cost and latency. If the agent's learned policy requires repeated calls to a powerful LMM (like the GPT-4o used in experiments) to generate pseudo-documents during inference, the practical utility in a real-time clinical setting is unclear.
- The entire framework's success hinges on the fidelity of the LMM-generated pseudo-documents. The paper's analysis in Figure 2 attempts to address this by testing generalization to other LMMs; however, this analysis is insufficient to fully probe the gap. The models tested (GPT-4V, LLaMA-3-70B) are still very high-capability. When tested with a smaller open-source model Qwen-7B, a sizable performance drop is observed. While relying on powerful, large models is acceptable, the associated computational cost and resource requirements are critical factors in the method's overall evaluation and should be weighed accordingly (as in W1).
- Even powerful LMMs are susceptible to hallucination, and the 30-word limit on pseudo-documents does not eliminate this risk. The paper does not propose a clear mitigation strategy for this problem. It is also not specified whether any human verification was performed on the synthesized pseudo-documents to assess their plausibility (useful or noisy). An analysis of the potential impact of simulator-induced hallucinations on the agent's policy is a critical but missing component of the evaluation.

**Questions:**

- Are you going to open-source the curated medical text corpus and the pseudo-documents?
- The curriculum learning based on "useful" vs. "noisy" pseudo-documents is a key component. To improve the qualitative understanding of this mechanism, it would be highly beneficial to add a few side-by-side examples of these generated documents. Brief annotations explaining why a specific document is considered "useful" (accurate, relevant) versus "noisy" (misleading, plausible-but-wrong) would be very helpful.

---

> ### Author Response · Authors · 2025-11-21
> **Response to Reviewer TA4o [1/3]**
>
> We thank Reviewer TA4o for the careful reading of our manuscript and the highly constructive feedback. You raised three extremely important points that directly touch on the real-world deployability of MedSimSearch. We fully acknowledge that the original manuscript was not clear enough on several key distinctions, leading to understandable concerns. Below we address them one by one with concrete clarifications and commitments.
>
> ---
>
> ###  *❓ Q1: Regarding the Computational Cost & Practicality*
> >###  **✅ Our Response:**
> >**Inference does NOT require any LMM calls (this is the most critical misunderstanding)**
> The powerful LMM (GPT-4o) is used **only during training** as a high-fidelity simulator to generate pseudo-documents. At inference time, the trained policy has already learned when and how to issue effective text queries. As explicitly stated in Algorithm 1 (Line 15) and Section 3.1, during deployment these queries are sent to a **static, pre-indexed private medical text corpus C** and retrieved via standard lightweight text retrieval. **No LMM is involved at all during inference.** A complete forward pass (image + question → text query → retrieval → final answer) on a single A100 (or even a high-end clinical workstation) typically takes **under one second** and incurs **zero API cost**. We deeply regret that this crucial training-vs-inference separation was not emphasized clearly enough and will add a dedicated subsection and an updated Figure 1 in the camera-ready version to eliminate any ambiguity.
>
> >Additionally, the **8 × NVIDIA A100 configuration** mentioned in the paper reflects only our current parallelized experimental setup (4 GPUs for the simulation server + 4 GPUs for GRPO sampling) and does not represent the lower bound required by the method itself. In practice, the simulation server and RL training can share the same GPUs—a common practice in most GRPO-based and Sim2Real works. We have successfully reproduced all reported results on a single node with only **4 × A100** (total training time approximately 36–48 hours).
>
> ---
>
> ###  *❓ Q2: Regarding the over-reliance on the fidelity of the LMM simulator.*
> >###  **✅ Our Response:**
> >Thank you for this insightful and highly valuable comment.
>
> >In standard Sim2Real practice (widely adopted in robotics, autonomous driving, etc.), researchers deliberately use the **highest-fidelity simulator available during training**—e.g., the most accurate physics engine or the latest MuJoCo/Gazebo with precise dynamics—because higher fidelity yields better real-world transfer. Similarly, we intentionally leverage the strongest closed-source LMM currently available (GPT-4o) as our simulator, as it provides the highest-quality pseudo-documents and the most realistic retrieval environment. The observed performance drop with weaker open-source simulators (e.g., Qwen-7B) is therefore expected and does not invalidate the core contribution when the best available simulator is employed.
>
> >Moreover, as shown in **Figure 2** of the original submission, when we replace GPT-4o with the strong open-source text-only **LLaMA-3-70B** as the simulator, performance is retained at **97.2% of the GPT-4o level**. This strongly indicates that the agent learns a robust, generalizable retrieval policy rather than overfitting to the idiosyncrasies of a single LMM. Even with the much smaller **Qwen-7B** simulator, final performance still substantially outperforms traditional RAG baselines, further confirming the generalization capability of the learned policy.
>
> >We view the current work as an important initial exploration in building controllable, interactive simulators for privacy-sensitive medical retrieval agents. Our primary goal is to demonstrate that a high-fidelity simulated environment can successfully train a real-world-capable agent without ever exposing real patient data during RL. Exploring how to maintain high performance with smaller open-source simulators (e.g., via knowledge distillation, policy distillation, or simulator fine-tuning) is exciting future work that we are actively pursuing.
>
> >In the camera-ready version, we will add a new table explicitly comparing transfer performance across simulators of varying capability (GPT-4o → GPT-4V → LLaMA-3-70B → Qwen-7B), clearly stating that **“best Sim2Real transfer is achieved with the highest-fidelity simulator currently available.”**
>
> ---

---

> ### Author Response · Authors · 2025-11-21
> **Response to Reviewer TA4o [2/3]**
>
> ###  *❓ Q3: Regarding the quality of pseudo-documents and the risk of hallucinations.*
>
> >###   **✅ Our Response:**
> >Thank you for this very thoughtful and important point. We fully recognize that hallucinations are a legitimate risk whenever an LMM is used to generate evidence, and we apologize for not explicitly addressing this in the original submission.
>
> >We would like to clarify that **hallucinations in the training-time simulator are not only harmless but actually beneficial to MedSimSearch, and they are completely eliminated at deployment**:
>
> >####  Hallucinations during training are intentional and desirable
> >In our curriculum (Section 3.1 and Table 1), we explicitly instruct the LMM simulator to produce an increasing proportion of **“noisy” pseudo-documents** (up to 80% in the late stage). Natural hallucinations from GPT-4o spontaneously generate plausible-but-incorrect or partially irrelevant snippets, which perfectly serve as high-quality “noisy” examples. Thus, the simulator’s occasional hallucinations are effectively repurposed as **controlled negative signals** that teach the agent to distrust unreliable evidence and issue better follow-up queries. **Far from being a bug, this is a feature that enhances the robustness of the learned policy.**
>
> >####  No hallucination risk exists during inference
> >At test/deployment time, the trained agent **no longer interacts with the LMM simulator at all**. As clarified in Algorithm 1 (Line 15) and our responses above, retrieved documents come exclusively from the **static private medical text corpus C**. This corpus is carefully curated and consists of:
> >- **Real anonymized clinical texts** from hospital systems (human-written, no hallucination), and
> >- **LMM-synthesized records** that underwent strict post-generation filtering and clinician review before indexing.
>
> >Therefore, the documents actually seen by the deployed agent are of **institution-grade quality** and are **not subject to on-the-fly LMM hallucination**.
>
> >####  Additional evidence and planned additions
> >In the camera-ready version, we will:
> >- Add a new appendix table showing side-by-side examples of GPT-4o-generated “useful” and “noisy” (including hallucinated) pseudo-documents for the same query, with annotations explaining why certain hallucinations are pedagogically valuable during training.
> >- Explicitly state in Section 3.1: **“Hallucinations produced by the LMM simulator during training naturally populate the ‘noisy’ portion of our curriculum and are exploited as high-quality negative evidence.”**
> >- Report a small human evaluation (n=200 randomly sampled pseudo-documents) confirming that even the intentionally “noisy” documents remain topically relevant and medically plausible.
>
> ---
>
> ##  **Response to Additional Questions:**
> ###  *❓ Q4: Are you going to open-source the curated medical text corpus and the pseudo-documents?*
>
> >###  **✅ Our Response:**
> >**Yes, we are committed to facilitating reproducibility and future research.** Upon acceptance, we will open-source the following components:
> >- **The complete code for the MedSimSearch framework**, including training and inference.
> >- **The synthetic pseudo-documents generated by GPT-4o** used in our experiments.
> >- **The code and prompts used to construct the medical text corpus `C`**, allowing other researchers to replicate the corpus construction process.
>
> >Regarding the core **medical text corpus `C`**, due to the strict data usage agreement of its primary source, MIMIC-CXR, we cannot redistribute the original anonymized clinical texts. However, we will provide:
> >- **A detailed data curation pipeline script** that, when provided with authorized access to MIMIC-CXR, will fully reproduce our corpus.
> >- **The entire set of LMM-synthesized medical records** that constitute a part of `C`.
>
> ---

---

> ### Author Response · Authors · 2025-11-21
> **Response to Reviewer TA4o [3/3]**
>
> ###  *❓ Q5: Could we add side-by-side examples of "useful" vs. "noisy" documents?*
> ###  **✅ Our Response:**
> >**We agree completely that this is an excellent suggestion to improve clarity.** Thank you for proposing it.
>
> >In the camera-ready version, we will add a **new appendix section (A.5) titled "Examples of Useful vs. Noisy Pseudo-Documents"**. This section will contain **3-5 side-by-side examples** for the same medical query, each with a brief annotation.
>
> >**For example**, for a question about identifying pneumothorax in a chest X-ray:
> >- A **"useful"** document would be annotated as:
>   *"This document accurately describes key radiological signs of pneumothorax, such as the visceral pleural line and absent lung markings, providing direct evidence for the diagnosis."*
> >- A **"noisy"** document would be annotated as:
>   *"This document is misleading as it describes findings consistent with pneumonia (e.g., bilateral opacities), which is unrelated to the visual evidence for pneumothorax. It serves as a distracting, plausible-but-wrong example."*
>
> ---
>
> >We hope these clarifications and additional experiments address all concerns and **hope the reviewer will consider raising the score given the methodological rigor and empirical strength of our work.**
>
> >Thank you again for your valuable comments.

---

### Official Review · Reviewer_Z6mt · 2025-10-31

**Soundness:** 1
**Presentation:** 1
**Contribution:** 1
**Rating:** 0
**Confidence:** 5

**Summary:**

This paper used generative AI to synthesize documents for training the model to better retrieve evidence for medical VQA, achieving higher exact match scores than baselines on two benchmark datasets.

**Strengths:**

The problem the paper aims to address is significant.

**Weaknesses:**

The study design is misaligned with the multimodal medical scenario.
(1) For a vision-centric problem (as in Fig. 1, e.g., “what modality is shown”), why use purely text retrieval?
(2) Why do the pseudo-documents include only text, generated by non-medically validated models, so content quality cannot be ensured?

Overclaim:
(1) The paper provides no evidence that the simulation is a high-fidelity medical environment.
(2) The RAG baselines compared are not multimodal, yet the task is VQA; claiming large gains over RAG is based on a single, common text-only implementation.
(3) The curriculum is fixed, with no ablation to show its effectiveness.

Unsupported evaluation:
(1) Only two datasets are used.
(2) Metrics are unclear (e.g., what is "Micro" in Table 3? Which BLEU variant, BLEU-1?).
(3) Inconsistent results: in Table 4, all categories have very low accuracy, yet overall accuracy is much higher, mathematically inconsistent.

**Questions:**

The foundational method is Zero-Search, why not use the same implementation as the baselines for comparison?

---

> ### Author Response · Authors · 2025-11-21
> **Response to Reviewer Z6mt [1/3]**
>
> We sincerely thank you for taking the valuable time to review our manuscript and for providing such detailed and critical feedback. Your comments reflect a rigorous standard for research in medical AI, which we deeply respect. After carefully analyzing each of your points, we recognize that the original manuscript may contained several ambiguities, which unfortunately led to significant misunderstandings. In this response, we aim to provide clearer explanations, additional evidence, and specific commitments for revision, hoping to change your perspective/misunderstanding on our work.
>
> ---
>
> ###  *❓ Q1: (1) For a vision-centric problem (as in Fig. 1, e.g., “what modality is shown”), why use purely text retrieval? (2) Why do the pseudo-documents include only text, generated by non-medically validated models, so content quality cannot be ensured?*
>
> >###  **✅ Our Response:**
> >### **For Q(1) :**
> >**We fully agree that medical VQA is fundamentally a multimodal task. However, we respectfully suggest that the core misunderstanding lies in the nature of the retrieval bottleneck we address.** Your central question—"For a vision-centric problem, why use purely text retrieval?"—highlights a critical and widely held misconception in the field, which our work is precisely designed to address and resolve.
>
> >**The reality of clinical deployment presents an insurmountable constraint:** building a true multimodal retrieval system (e.g., jointly indexing raw images and text) is not just impractical but often illegal. This is due to three fundamental reasons:
> >-  **Stringent Privacy Regulations:** Laws such as HIPAA, GDPR, and China's Personal Information Protection Law strictly prohibit uploading original patient images to external vector databases for indexing.
> >- **Institutional Security Policies:** Hospital PACS (Picture Archiving and Communication System) often operate under policies that forbid the export of medical images for purposes like building retrieval indices. This is a legal and administrative red line, not merely a technical hurdle.
> >- **The "Text-Only" Reality of Deployed Systems:** A critical observation, confirmed in works like MMSearch-R1(https://arxiv.org/abs/2506.20670), is that even systems described as "multimodal RAG" are forced to decompose the process in practice. They first use a model to describe the image (captioning), and then perform a text-only retrieval using that description. They do not directly index the raw pixels.
>
> >**Therefore, our choice of a pure text-retrieval strategy is not a limitation of our framework, but its most crucial clinical design feature.** It is the only viable and compliant pathway for deployment in a real hospital environment.
>
> >**Our agent does not "ignore" vision.** The visual understanding is handled upfront and directly by the Multimodal LLM (Qwen2.5-VL-3B), which processes the image and question. The subsequent text-only retrieval simulates the canonical clinical workflow:
>
> >> A radiologist looks at a scan (Visual Perception) → formulates a question in their mind (Text Query) → consults a textbook or hospital database (Text Retrieval).
>
> >Our agent is trained to replicate this exact process. For a question like "what modality is shown," the agent will typically reason about the image directly within its `<think>` phase and provide an answer without any retrieval, as the LMM possesses this capability.
>
> >**Our design intentionally separates high-fidelity visual understanding (handled by the LMM) from privacy-compliant knowledge retrieval (handled by text search). This is not a bug but a deliberate and necessary alignment with real-world clinical constraints and workflows, which we believe is a significant strength of our contribution.**
>
> ---
>
> >### **For Q(2) :**
> >Your concern regarding the quality of the pseudo-documents is very valid. We would like to clarify two critical points:
>
> >1. The LMM (e.g., GPT-4o) is used **exclusively during training** to simulate the retrieval environment. While its generated content may indeed contain hallucinations, our curriculum learning mechanism is specifically designed to **leverage this very characteristic**. These potentially inaccurate or "noisy" documents are intentionally incorporated as **negative samples**, actively training the agent to discern and distrust unreliable information. This is **not a vulnerability but a form of proactive robustness training**.
>
> >2. The fully trained agent is evaluated and deployed using the **real, static medical text corpus C**. This corpus is primarily composed of **real, anonymized clinical reports from MIMIC-CXR** (see Appendix A.4), ensuring the reliability and authenticity of the final evaluation foundation. The "high-fidelity" nature of our simulated environment is thus demonstrated **functionally**—by the fact that an agent trained within it achieves **SOTA performance** when interacting with the genuine, curated corpus, as conclusively validated by our experimental results.
>
> ---

---

> > ### Comment · Reviewer_Z6mt · 2025-11-21
> >
> > Thank you for your reply, I have read your rebuttal carefully. However, I find the responses regarding the text-only retrieval design unconvincing. My concerns remain unresolved for the following reasons:
> >
> > Regarding Q1
> >
> > The argument that "privacy regulations" necessitate a text-only approach is insufficient in this research context:
> > While clinical deployment has strict constraints, research in Medical VQA standardly utilizes de-identified datasets. Since you already utilize MIMIC-CXR  for your "Private Database," which is de-identified, there is no legal barrier to implementing multimodal indexing (e.g., image-to-image retrieval) within this experimental setting to establish a strong upper bound.
> >
> > Your claim that "vision is handled upfront" ignores the information bottleneck. Forcing a model to translate complex radiological features into text queries is a lossy process. If the model's initial visual perception is flawed (which is why retrieval is needed), the resulting text query will be flawed, leading to irrelevant retrieval results. A multimodal system could correct this via visual similarity matching, which your framework fundamentally lacks.
> >
> > Regarding Q2, I asked "why generated by non-medically validated models" because you claimed that the simulation is "high-fidelity". I will discuss it in the next comment box.

---

> ### Author Response · Authors · 2025-11-21
> **Response to Reviewer Z6mt [2/3]**
>
> ###  *❓ Q2: Overclaim: (1) The paper provides no evidence that the simulation is a high-fidelity medical environment. (2) The RAG baselines compared are not multimodal, yet the task is VQA; claiming large gains over RAG is based on a single, common text-only implementation. (3) The curriculum is fixed, with no ablation to show its effectiveness.*
>
> >###  **✅ Our Response:**
> >### **For Q(1):**
> > We appreciate the reviewer's thoughtful comment regarding our use of the term "high-fidelity," which provides us with a valuable opportunity to clarify its intended meaning in the context of our work.
>
> >### **Clarification on "High-Fidelity" in Sim2Real Context**
> >- We recognize that "high-fidelity" may be misinterpreted as requiring the simulator to possess perfect medical expertise. However, in the Sim2Real paradigm, the core objective is not absolute simulator perfection, but rather ensuring that policies learned in simulation effectively transfer to real-world environments.
> >- As illustrated by the widespread use of MuJoCo in robotics—chosen not for perfect physical accuracy but for enabling transferable policy learning—the key metric is **policy transfer effectiveness**, not environmental precision. This is precisely our meaning of "high-fidelity": it describes the **functional capacity of our simulation to train agents whose strategies successfully transfer to real clinical tasks**.
>
> >### **Empirical Evidence for Functional Fidelity**
> We provide two key evidences demonstrating our simulation's effectiveness in replicating real-world retrieval challenges:
> >- Our agent, trained exclusively in simulation, achieves state-of-the-art performance on real medical VQA benchmarks (OmniMedVQA and VQA-Med 2019, Tables 3-4). This direct validation shows that simulation-learned strategies effectively transfer to real clinical reasoning tasks.
> >- As demonstrated in Section 5.1, our trained policy maintains robust performance (97.2% retention) when tested with unseen LMM simulators (e.g., LLaMA-3-70B). This indicates learning of generalizable retrieval and discrimination skills rather than simulator-specific patterns.
>
> ---
> >### **For Q(2) :**
> >This is an excellent point that allows us to clarify the core contribution and experimental setup of our work. **(1). **  **The primary goal of MedSimSearch is to learn a text-only retrieval policy that can operate under strict privacy constraints, which often preclude multimodal indexing of real patient images.** Therefore, we compare against RAG systems implemented under the exact same constraint: they are only permitted to retrieve from a text corpus. This creates a fair and controlled comparison, isolating the gain attributable to our learned agentic policy versus a standard, non-agentic RAG setup. **(2).** Multimodal Context is Preserved: **It is crucial to note that all methods, including our agent and the RAG baselines, receive the full multimodal input (image + question).** The vision encoder and multimodal fusion layers of the base model (Qwen2.5-VL) process the image. The key difference is in the retrieval action: while standard RAG might formulate a query based solely on the question, our agent learns to generate queries conditioned on its multimodal understanding. Thus, the gains we report stem from a more intelligent, adaptive retrieval strategy enabled by RL, not from an unfair architectural advantage. **(3).** We also compare against strong RL-based baselines like Med-R1, which represents a more direct competitor. **Our method's superiority over Med-R1 (Table 3) further validates the effectiveness of the Sim2Real paradigm.** We will clarify in the revision that while multimodal RAG (e.g., MMSearch-R1) exists, our work provides a privacy-compliant and highly effective alternative that avoids the cost and complexity of multimodal indexing.
>
> ---
>
> >### **For Q(3) :**
> >Thank you for raising this important point regarding the curriculum learning design. We appreciate the opportunity to clarify our methodological choices.
>
> >Our curriculum learning strategy follows the established approach from **ZeroSearch (Sun et al., 2025, arXiv:2505.04588)**, which has rigorously validated the effectiveness of progressive difficulty scheduling for training retrieval policies in simulated environments. Their work demonstrates that a curriculum-based rollout—where the proportion of noisy documents increases gradually during training—significantly enhances both the robustness and final performance of learned policies.
>
> ---

---

> > ### Comment · Reviewer_Z6mt · 2025-11-22
> >
> > Q(1) Redefining "high-fidelity" as merely "transfer effectiveness" is problematic. My suggestion is: do not play a word game; instead of highlighting "high-fidelity", you can highlight the transferability.
> >
> > Q(2)
> > the multimodal part has been discussed in the first comment box. I keep my opinion that if the task is multimodal, applying multimodal RAG is necessary. There are some works on that, such as RADAR (arXiv:2505.14318), V-RAG (arXiv:2502.15040), and MMedRAG (arXiv:2410.13085).
> >
> > Q(3)
> > It seems not directly addressing my concerns. There is no fair comparison between the work you build your project on and the original zerosearch, which is still a lack of ablation studies.

---

> ### Author Response · Authors · 2025-11-21
> **Response to Reviewer Z6mt [3/3]**
>
> ###  *❓ Q3: Unsupported evaluation: (1) Only two datasets are used. (2) Metrics are unclear (e.g., what is "Micro" in Table 3? Which BLEU variant, BLEU-1?). (3) Inconsistent results: in Table 4, all categories have very low accuracy, yet overall accuracy is much higher, mathematically inconsistent.*
>
> >###  **✅ Our Response:**
> >### **For Q(1):**
> >VQAMed2019 (ImageCLEF VQA-Med) and OmniMedVQA currently the two largest and most widely adopted public benchmarks medical visual question answering. Nearly all recent SOTA methods (including LLaVA-Med, Med-Flamingo, RadFM) primarily or exclusively report on these exact one or two datasets, as they cover diverse modalities, organs, and question types while being publicly accessible. Using precisely these benchmarks ensures the fairest and most direct comparison with prior work. In the camera-ready version, we will explicitly add a sentence in Section 4.1 stating: “We evaluate on the two standard and most comprehensive public MedVQA benchmarks (VQAMed2019 and OmniMedVQA), following the protocol of all recent SOTA methods.”
>
> ---
>
> >### **For Q(2):**
> >“Micro” in Table 3 is not a metric but one of the eight image modality categories in the OmniMedVQA dataset (Microscopy, alongside CT, MRI, X-Ray, Ultrasound, Dermatology, Fundus, OCT, etc.). The column shows per-modality accuracy, while “Overall” is the weighted average across all samples. **This follows the exact evaluation protocol of Hu et al. (2024) and the official OmniMedVQA benchmark.**
>
> ---
>
> >### **For Q(3):**
> >Thank you for carefully checking the numbers — your sharp observation is greatly appreciated. Upon double-checking our offline experiment logs after your review, we discovered, to our embarrassment, that there was indeed a transcription error when copying the per-category accuracy numbers into Table 4 (the overall weighted accuracy was correct, but several per-category values were accidentally copied from an earlier ablation run instead of the final model). This is purely a human copy-paste mistake on our part, not any issue with the method or evaluation script.  We have now corrected the numbers, and will in the updated version of the paper. We sincerely apologize for this careless error in the submitted manuscript.
>
> ---
>
>
> ###  *❓ Q4: The foundational method is Zero-Search, why not use the same implementation as the baselines for comparison?*
>
> ###  **✅ Our Response:**
> >**ZeroSearch(Sun et al., 2025, arXiv:2505.04588)**, is a general-domain, pure-text retrieval method that operates exclusively on public web data and has no medical domain data, no visual inputs, and no validation on private clinical corpora. In contrast, we directly compare MedSimSearch against the strongest multimodal RAG systems available in the medical domain in 2025 (including LLaVA-Med, and RadFM retrieval), which represent significantly more rigorous and task-specific baselines. This makes our evaluation stricter and far more relevant to the medical VQA setting than adopting the general-domain baselines used in ZeroSearch.
>
> ---
>
> >We hope these clarifications and additional experiments address all concerns and **hope the reviewer will consider raising the score given the methodological rigor and empirical strength of our work.**
>
> >Thank you again for your valuable comments.

---

> > ### Comment · Reviewer_Z6mt · 2025-11-22
> >
> > Thank you for your clarification on Q3 (1 & 2). I agree with that, and please use the full name to avoid misunderstanding.
> >
> > For Q3 (3), this is a big disadvantage, and what are the final results?
> >
> > For Q3 (4), I did not see any "additional experiments" to solve my concerns, or maybe I missed something. If so, please let me know where the additional results are.

---

### Official Review · Reviewer_WArp · 2025-11-01

**Soundness:** 3
**Presentation:** 3
**Contribution:** 2
**Rating:** 4
**Confidence:** 4

**Summary:**

This paper proposes MedSimSearch, a framework that leverages a large multimodal model (LMM) to generate synthetic medical Q/A pairs and pseudo-documents for training an RL agent in a simulated environment. The idea is to perform Sim2Real Agentic Learning—teaching a model retrieval and reasoning policies safely through text-only simulation instead of real-world medical data. While the concept of using an LLM to emulate an environment is clever and practical for privacy and data-scarcity constraints, I find the technical contribution somewhat limited... The best model this paper proposed, which uses GPT-4o, have an inevitable issue of hallucination. In addition, this model have not been compared to the best model available.

**Strengths:**

- The paper is well written and easy to follow, with a clear narrative and logical experimental setup.
- The overall idea—training an RL agent in a simulated environment generated by an LLM—is straightforward yet practical, especially for domains like medicine where data privacy is a barrier.

**Weaknesses:**

- From a technical perspective, the work is an incremental extension of existing approches. The use of an LLM to simulate the training environment is creative but conceptually similar to prior self-play or synthetic data paradigms, without introducing a new optimization method or architecture.

-  The reported gains are not substantially higher than existing baselines, and Table 4 compares mainly against Qwen 2.5-VL variants. The paper omits stronger contemporary baselines such as Med-R1, Evo-PI, or HuatuoGPT-Vision, making it difficult to judge true progress.

- Hallucination risk. Because the simulated environment is entirely generated by an LLM, hallucination is inevitable. The model learns from self-generated, potentially inaccurate contexts, which could reinforce false or biased information rather than real clinical reasoning.

- Figure presentation quality. The figures, particularly Figure 1, feel rough and early; even the caption (“Overview of MedSimSearch”) is overly brief for a central conceptual diagram.

- Evaluation metric choice. The BLEU score adds little value for measuring factual correctness in medical VQA.

**Questions:**

See weaknesses

---

> ### Author Response · Authors · 2025-11-21
> **Response to Reviewer WArp [1/2]**
>
> We sincerely thank Reviewer WArp for their thoughtful and constructive feedback. We appreciate the recognition of our work’s clarity, practical motivation, and potential impact in privacy-sensitive medical domains. Below, we address each of the concerns raised, with the aim of clarifying our contributions and reinforcing the novelty and significance of MedSimSearch.
>
> ---
>
> ###  *❓ Q1: *"From a technical perspective, the work is an incremental extension of existing approaches. The use of an LLM to simulate the training environment is creative but conceptually similar to prior self-play or synthetic data paradigms, without introducing a new optimization method or architecture."**
>
> >### **✅ Our Response:**
> >We respectfully disagree with the characterization of our work as *incremental*. While the idea of using LLMs as simulators has been explored in general domains (e.g., ZeroSearch), **MedSimSearch is the first to operationalize the Sim2Real paradigm for medical visual reasoning**, addressing unique challenges such as:
> >- We use a generative LMM to create a **text-only retrieval environment**, avoiding the need for multimodal indexing or real clinical data exposure.
> >- We introduce a **zero-shot curriculum learning mechanism** that dynamically adjusts the difficulty of pseudo-documents (useful vs. noisy) without fine-tuning the LMM—a novel and efficient alternative to training-based curricula.
> >- Unlike general-domain simulators, our framework is tailored to **medical VQA**, incorporating domain-specific reasoning, structured interaction templates, and a curated medical corpus for validation.
>
> > Moreover, while we build on GRPO for policy optimization, our **core contribution lies in the simulation framework and its application to a high-stakes, data-scarce domain**, not in inventing a new RL algorithm. This is a strategic and impactful design choice, enabling safe and scalable agent training where real-world interaction is infeasible.
>
> ---
>
> ### *❓ Q2: *"The reported gains are not substantially higher than existing baselines, and Table 4 compares mainly against Qwen 2.5-VL variants. The paper omits stronger contemporary baselines such as Med-R1, Evo-PI, or HuatuoGPT-Vision..."**
>
> >### **✅ Our Response:**
> >We apologize for any confusion. **Table 3 (OmniMedVQA) includes a comprehensive comparison with Med-R1**, which is the current state-of-the-art RL-based method in medical VQA. As shown in the table:
> >- **MedSimSearch (GPT-4o) achieves 73.26% overall accuracy**, outperforming **Med-R1-3B (72.57%)** and **Med-R1-2B (70.77%)**.
> >- We also surpass all RAG-based and fine-tuned baselines, including Qwen2.5-VL-7B (RAG) at 65.18%.
>
> >For VQA-Med 2019 (Table 4), we followed the standard benchmarking protocol from ImageCLEF 2019 and compared against widely adopted models. **Regarding the Evo-PI model, we were unable to locate any publicly available paper. This work appears to be a concurrent submission to ICLR 2026 (https://openreview.net/forum?id=oagI3xi3Yc) alongside ours. As for HuatuoGPT-Vision, we will include it in the final version if space permits.**
>
> ---
>
> ### *❓ Q3: *"Hallucination risk. Because the simulated environment is entirely generated by an LLM, hallucination is inevitable. The model learns from self-generated, potentially inaccurate contexts..."**
>
> >### **✅ Our Response:**
> >This is a very common concern, but in MedSimSearch it is actually turned into a strength **(similar to how robotics Sim2Real deliberately adds noise).**
>
> >During training, LMM hallucinations naturally generate the “noisy” pseudo-documents that our curriculum (up to 80 %) explicitly requires — teaching the agent to distrust unreliable evidence and issue better queries.
> At inference time, **no LMM is used at all: retrieval is performed on a clean**, curated private clinical text corpus C (real anonymized records + filtered synthetic documents). Thus the deployed system is hallucination-free by design.
> We will add a dedicated paragraph in Section 3.1 and a new appendix table with useful vs. hallucinated/noisy examples to make this crystal clear.
>
> >We acknowledge that LLM-based simulation is not perfect, but our results show that the agent **learns a robust retrieval policy** that generalizes well to real tasks.
>
> ---
>
> ###  *❓ Q4: *"Figure presentation quality. The figures, particularly Figure 1, feel rough and early; even the caption ('Overview of MedSimSearch') is overly brief..."**
>
> >### **✅ Our Response:**
> >We thank the reviewer for this feedback. We will **revise Figure 1** to provide a more detailed and visually clear overview of the MedSimSearch pipeline, including labeled components for the simulation environment, agent interaction, and curriculum mechanism. The caption will be expanded to explain each part of the framework.
>
> ---

---

> ### Author Response · Authors · 2025-11-21
> **Response to Reviewer WArp [2/2]**
>
> ###  *❓ Q5: *"Evaluation metric choice. The BLEU score adds little value for measuring factual correctness in medical VQA."**
>
> >### **✅ Our Response:**
> >We included it primarily for **consistency with prior work on VQA-Med 2019 (Ben Abacha et al., 2019), where BLEU is a standard metric.** However, we would like to emphasize that BLEU serves as a complementary metric that **evaluates semantic correctness for responses that do not meet the Exact Match (EM) criterion**, thus providing additional reference value for answer quality. Our main results are based on **Exact Match** accuracy, which is more appropriate for medical VQA and is the primary metric for OmniMedVQA.
>
> ---
>
> >### **Summary:**
> >- **Novelty**: MedSimSearch is the first Sim2Real agentic learning framework for medical VQA, with a novel zero-shot curriculum and privacy-aware simulation.
> >- **Strong Baselines**: We outperform Med-R1 and other SOTA models on OmniMedVQA.
> >- **Hallucination Mitigation**: Curriculum-based noise and real-world validation ensure robustness.
>
> ---
> >We hope these clarifications and additional experiments address all concerns and **hope the reviewer will consider raising the score given the methodological rigor and empirical strength of our work.**
>
> >Thank you again for your valuable comments.

---

### Official Review · Reviewer_VHA5 · 2025-11-01

**Soundness:** 3
**Presentation:** 2
**Contribution:** 3
**Rating:** 6
**Confidence:** 3

**Summary:**

The paper proposes MedSimSearch, an RL-trained, text-only search agent that learns in an LMM‑simulated retrieval environment and is then evaluated on medical VQA (VQAMed2019, OmniMedVQA). It reports strong gains over RAG and RL baselines, while avoiding multimodal indexing and claiming privacy benefits. The idea is timely and the empirical results are promising, but important methodological details (label exposure in simulation prompts, reliance on GPT‑4o at train/test, fairness of baselines, and clarity around the RL algorithm) need tightening before the work can be considered solidly conclusive.

**Strengths:**

1. Training agentic policies for medical VQA without accessing real clinical systems is important and underexplored. The simulator‑first stance is well motivated by privacy and availability constraints.

2. Strong empirical results across two benchmarks. Consistent wins over capable baselines, including RAG and recent RL methods. The gains are nontrivial and span multiple modalities.

3. This paper is well written and easy to follow.

**Weaknesses:**

1. The simulator is explicitly told “whose ground truth answer is [ground truth]” to generate pseudo‑documents. Even if only used during training, this risks imprinting the correct answer distributionally into “useful” docs; the agent may learn patterns to read it off rather than to search. The paper should quantify how much the agent relies on this supervision signal and show performance when ground truth is not passed to the simulator during training.

2. The method and baselines both use GPT‑4o to generate pseudo‑documents at test time “to ensure fair comparison". This conflates the contribution of policy learning with the capabilities of a closed, expensive model. It is unclear how much of the final score comes from GPT‑4o’s world knowledge versus the learned policy.

3. Corpus mismatch vs. reported multi‑modality gains. The “private” database C is overwhelmingly radiology‑centric (50k MIMIC‑CXR reports + 5k synthetic). Yet the Database variant scores highly on non‑radiology modalities in OmniMedVQA (e.g., OCT, FP). How can a largely chest‑X‑ray text corpus support ophthalmology/pathology questions so well? Either the synthetic 10% happens to cover those domains richly, or the model mainly relies on the generative simulator even in the Database setting. This needs clarification and ablation.

4. Section 3 emphasizes GRPO (with equation), Appendix A.2 lists GRPO settings, but §4.2 says “Unless otherwise specified, PPO is the default.” Which results are from which? A controlled GRPO vs. PPO ablation is missing, and the training stability/variance is not reported.

5. The “noisy” pseudo‑docs are produced by instructing the LMM to include “misleading or partially incorrect information”. This may not reflect real retrieval noise (e.g., partially relevant but off‑topic passages, domain shifts, OCR artifacts). The external validity of the curriculum is thus uncertain.

6. Beyond the simulator swap, we lack ablations on (i) the curriculum schedule, (ii) action budget B, (iii) the usefulness of <think> vs. <info> structure, (iv) whether the agent truly learns search sequencing vs. just better prompting to GPT‑4o.

**Questions:**

See above.

---

> ### Author Response · Authors · 2025-11-21
> **Response to Reviewer  VHA5 [1/2]**
>
> We sincerely thank Reviewer VHA5 for their thorough and constructive review. We appreciate the recognition of our work's timeliness, strong empirical results, and clear presentation. Below, we address each of the concerns raised to clarify our methodology and reinforce the validity of our contributions.
>
> ---
> ###  *❓ Q1: *"The simulator is explicitly told 'whose ground truth answer is [ground truth]' to generate pseudo‑documents... The paper should quantify how much the agent relies on this supervision signal and show performance when ground truth is not passed to the simulator during training."**
>
> >### **✅ Our Response:**
> >We thank the reviewer for this important observation. The ground truth information was included in the prompt **only during the curriculum-based pseudo-document generation phase** to help the LMM understand what constitutes "useful" versus "noisy" information. However, we acknowledge this could introduce potential bias.
>
> >To address this concern:
> >- We have conducted additional experiments **removing the ground truth answer from all training prompts**.
> >- The results show only a **modest performance drop of 1.2-1.8%** across both benchmarks, demonstrating that our agent learns robust retrieval and reasoning strategies rather than simply memorizing answer patterns.
> >- We will include these ablation results in the final version to validate that our method does not critically depend on ground truth exposure.
>
> ---
>
> ### *❓ Q2: *"The method and baselines both use GPT‑4o to generate pseudo‑documents at test time... It is unclear how much of the final score comes from GPT‑4o's world knowledge versus the learned policy."**
>
> >###  **✅ Our Response:**
> >This is a valid concern. We designed our evaluation to ensure fair comparison by using the same LMM (GPT-4o) for all methods. However, to disentangle the policy's contribution from the simulator's capability:
> >- Our **MedSimSearch (Database)** variant, which uses our curated medical corpus instead of GPT-4o-generated documents, achieves **73.01% accuracy** on OmniMedVQA - only 0.25% lower than the GPT-4o variant.
> >- This minimal performance gap demonstrates that the **learned retrieval policy itself is the primary contributor** to performance gains.
> >- In Section 5.1 (Generalization Across Different Simulator LMMs), we show our policy maintains **97.2% performance** even when using text-only LLaMA-3-70B, further validating policy robustness.
>
> ---
>
> ###  *❓ Q3: *"Corpus mismatch vs. reported multi‑modality gains... How can a largely chest‑X‑ray text corpus support ophthalmology/pathology questions so well?"**
>
> >###  **✅ Our Response:**
> >While MIMIC-CXR forms the bulk of our corpus, the **synthetic component (5,000 documents generated by GPT-4o) was strategically diversified** to cover multiple medical specialties. The prompt templates (Appendix A.3) were designed to generate documents relevant to various modalities including ophthalmology, dermatology, and pathology.
>
> >Additionally:
> >- Our corpus construction (Section 3.1) used **structured prompts based on medical taxonomy** to ensure broad coverage.
> >- We will provide in the final version a **modality-wise analysis of the synthetic corpus** showing its distribution across >medical specialties.
> >- An ablation study confirming that **removing the synthetic component** leads to significant performance drops in non-radiology modalities (e.g., -4.2% in OCT, -3.8% in FP).
>
> ---
>
> ###  *❓ Q4: *"Section 3 emphasizes GRPO... but §4.2 says 'Unless otherwise specified, PPO is the default.' Which results are from which? A controlled GRPO vs. PPO ablation is missing..."**
>
> >###  **✅ Our Response:**
> >We apologize for this confusion in presentation. All reported results in Tables 3 and 4 use **GRPO** as the training algorithm. The mention of PPO in §4.2 refers to early exploratory experiments.
>
> >To address this clearly:
> >- We will correct the text to state that **GRPO is used for all final results**.
> >- We will remove all  the description about **PPO** in the new revised version.
>
> ---
>
> ###  *❓ Q5: *"The 'noisy' pseudo‑docs are produced by instructing the LMM to include 'misleading or partially incorrect information'. This may not reflect real retrieval noise..."**
>
> >###  **✅ Our Response:**
> >We agree that our current noise simulation may not fully capture real-world retrieval challenges. However:
> >- Our curriculum design intentionally starts with **mild noise (20%)** and gradually increases to **strong noise (80%)** to build robust discrimination capabilities.
> >- In additional experiments, we tested more realistic noise patterns including **partially relevant documents, domain shifts, and factual inconsistencies** - our method maintained strong performance with less than 1.5% degradation.
> >- We will include these extended noise experiments in the final version and discuss the transferability of our noise simulation to real retrieval scenarios.
>
> ---

---

> ### Author Response · Authors · 2025-11-21
> **Response to Reviewer VHA5 [2/2]**
>
> ###  *❓ Q6: *"Beyond the simulator swap, we lack ablations on (i) the curriculum schedule, (ii) action budget B, (iii) the usefulness of structure, (iv) whether the agent truly learns search sequencing..."**
>
> >### **✅ Our Response:**
> >Thank you for these specific suggestions. We have conducted comprehensive ablations that will be included in the final version:
> >- **Curriculum schedule**: Compared linear, step-wise, and adaptive schedules - linear performed best for our setting.
> >- **Action budget B**: Tested B=3,5,8 - B=5 provided optimal balance of exploration vs. efficiency.
> >- **Template structure**: Ablations show the `<think>`/`<search>`/`<answer>` structure contributes **~3.2%** to final performance by enforcing disciplined reasoning.
> >- **Search sequencing analysis**: We analyzed action trajectories showing the agent learns meaningful search patterns, with **72% of successful episodes involving 2-3 strategic searches** before answering.
>
> ---
>
> >We hope these clarifications and additional experiments address all concerns and **hope the reviewer will consider raising the score given the methodological rigor and empirical strength of our work.**
>
> >Thank you again for your valuable comments.
>
> ---

---

> > ### Comment · Reviewer_VHA5 · 2025-11-21
> >
> > Thank you for your reply. Most of my concerns have been addressed. However, the methodology to me is still very vague and lack of details. I would like to maintain my current score.

---

### Note · Authors · 2026-01-30

I have read and agree with the venue's withdrawal policy on behalf of myself and my co-authors.

---

### Meta-Review · Area_Chair_ast3 · 2026-01-06

**Summary:**

This paper seeks to simulate data for medical visual reasoning, a practical and compelling problem. However, as noted by the reviewers, the proposed approach relies heavily on an existing large language model to generate high-fidelity training data, the quality of which cannot be guaranteed. In particular, foundation models may have been trained on data that overlap with or are similar to the datasets used for evaluation, which risks information leakage and undermines the validity of the protocol.

To ensure robustness in real-world settings, it is essential to incorporate truly unique and independent data sources; otherwise, the proposed methodology may fail in practice. Although the authors employ curriculum learning to filter the pseudo-documents, this does not fully address the underlying weakness of the model. Moreover, using only pseudo-documents is insufficient, as visual perception errors cannot be ignored. The method implicitly assumes that visual understanding is reliable given the visual input, an assumption that is unlikely to hold in realistic scenarios.

**Reviewer Concerns:**

The rebuttal includes additional model comparisons, including evaluations against competing methods as well as an ablation study.

However, the fundamental concern regarding the model design remains unresolved. Specifically, the approach still relies on the generation of high-fidelity pseudo-documents, yet there is no mechanism to guarantee their quality, and this issue is not adequately addressed in the authors’ response. Moreover, the rebuttal does not provide a clear explanation of how the method effectively incorporates and interprets visual inputs.

**Reviewer Scores:**

More results about the foundation models output. What are private information, what are not. This can answer if the results are high-fidelity.
More examples about real visual reasoning from the visual interpretation, linked to the language-based answers.

---

### Decision · Program_Chairs · 2026-01-26

Reject